# Facile Synthesis of Nitrogen-Doped Carbon Quantum Dots with Chitosan for Fluorescent Detection of Fe^3+^

**DOI:** 10.3390/polym11111731

**Published:** 2019-10-23

**Authors:** Li Zhao, Yesheng Wang, Xihui Zhao, Yujia Deng, Yanzhi Xia

**Affiliations:** 1School of Chemistry and Chemical Engineering, Qingdao University, Qingdao 266071, China; 2State Key Laboratory of Bio-fibers and Eco-textiles, Shandong Collaborative Innovation Center of Marine Biobased Fibers and Ecological Textiles, Institute of Marine Biobased Materials, Qingdao University, Qingdao 266071, China

**Keywords:** chitosan, N-CQDs, Fe^3+^ detection, fluorescent sensor

## Abstract

A facile, economical, and one-step hydrothermal method was used to prepare highly luminescent nitrogen-doped carbon quantum dots (N-CQDs) with chitosan as both carbon and nitrogen sources. The as-prepared N-CQDs have an average size of 2 nm and exhibit excitation wavelength-dependent fluorescence with a maximum excitation and emission at 330 and 410 nm, respectively. Furthermore, due to the effective quenching effect of Fe^3+^ ions, the prepared N-CQDs can be used as a fluorescent sensor for Fe^3+^ ion-sensitive detection with a detection limit of 0.15 μM. The selectivity experiments revealed that the fluorescent sensor is specific to Fe^3+^ even with interference by high concentrations of other metal ions. Most importantly, the N-CQD-based Fe^3+^ ion sensor can be successfully applied to the determination of Fe^3+^ in real water samples. With excellent sensitivity and selectivity, such stable and cheap carbon materials are potentially suitable for the monitoring of Fe^3+^ in environmental application.

## 1. Introduction

Carbon quantum dots (CQDs), a kind of novel zero-dimensional nanomaterial, have attracted great interest due to their unique optical properties, good water solubility, high stability, low toxicity, excellent biocompatibility, and low environmental impact [1]. This newly discovered material has a wide range of potential applications, including fluorescent inks [2], bioimaging [3], photocatalysis [4], medical diagnosis [5], sensing [6], and ion detection [7]. Since the discovery of carbon dots, various methods for synthesizing carbon dots have been discovered, such as arc discharge [8], laser ablation [9], electrochemical synthesis [10], and microwave heating [11], hydrothermal synthesis [12]. Among them, hydrothermal synthesis is considered to be the most accessible method because of its simple operating conditions, low energy consumption, and low-cost apparatus. Additionally, although the study on the preparation of CQDs has gradually matured, more and more researchers are paying attention to the preparation of CQDs by green, low-cost, natural precursors, especially waste biomass materials, because they have the advantages of low cytotoxicity, high biocompatibility, and lower price compared to traditional CQDs and organic fluorescent dyes.

The iron ion is one of the heavy metal ions that induce many environmental issues. After heavy metals are released into the environment, even at low concentrations, they can have serious effects on the surrounding environment and organisms [13]. Iron ions are one of the most abundant elements on earth and a necessary trace element for biological processes [14]. It plays an important role in environmental, industrial, human, and biological systems. As we all know, the Fe^3+^ ion is the basic element of the heme group and plays an important role in oxygen metabolism [15]. However, its deficiency, beyond the normal allowable limits, may interfere with the balance of cellular processes and may lead to serious diseases, such as organ failure, Alzheimer’s disease, inflammation, or even death. The U.S. Environmental Protection Agency (EPA) has set limits for iron in drinking water and food to 0.3 ppm (∼5.4 μM) [16]. Therefore, the development of an effective sensing system for selective and sensitive analysis of Fe^3+^ ions is of great significance for practical applications. To date, a variety of well-established methods for the detection of Fe^3+^ ions have been reported, such as inductively coupled plasma mass spectrometry (ICP-MS) [17], spectrophotometry [18], atomic absorption spectrometry (AAS) [19], and electrochemical methods [20]. However, these methods have their own limitations, for example, highly specialized instruments with complicated procedures that involve time-consuming analytical processes and high cost [21]. Due to the cheap equipment, simple operation, fast response, and high sensitivity, fluorescence spectroscopy has been widely used in the detection of iron ions [22].

Chitosan is a natural polysaccharide with excellent biocompatibility and biodegradability, consisting of β-(1-4)-linked D-glucosamine and N-acetyl-D-glucosamine, and is obtained by the partial deacetylation of chitin derived from crustacean shells [23,24]. Due to its unique properties such as biocompatibility, biodegradability, non-toxicity, antibacterial properties, heavy metal ion sequestration, and hydrophilicity, chitosan is widely used in biology, medicine, food, cosmetics, industry, and other fields [25,26,27]. Due to the large amount of amino groups (-NH_2_) and hydroxyl groups (-OH) present in the structure [28], it has natural chelating properties that can be combined and used to remove heavy metal ions from water. Therefore, there is growing interest to explore the use of chitosan in water treatment for the removal of heavy metals [29,30]. In this work, water-soluble nitrogen-doped carbon quantum dots (N-CQDs) were synthesized via the hydrothermal method with chitosan as carbon and nitrogen sources. The prepared N-CQDs could be used to selectively and sensitively detect Fe^3+^ in practical aqueous solutions.

## 2. Experimental Section

### 2.1. Materials

Chitosan (deacetylation degree 82.5%) was prepared in our lab with shrimp shell, which is usually discarded as shellfish waste in the fishing and seafood industries. The deacetylation degree was determined by the titration method [31]. All metal salts (CdCl_2_, MgCl_2_, FeCl_3_, FeCl_2_, CuCl_2_, Pb(NO_3_)_2_, Ni(NO_3_)_2_, AgNO_3_, ZnCl_2_, AlCl_3_, NaCl, Co(NO_3_)_2_, CaCl_2_, BaCl_2_, HgCl_2_) were purchased from Beijing Chemical Reagents Factory (Beijing, China). All chemicals were of analytical grade and used directly without further purification. Ultrapure water was used in all the experiments.

### 2.2. Synthesis of N-CQDs

Luminescent and biocompatible chitosan-based N-CQDs were prepared by a facile hydrothermal synthetic strategy. The detailed preparation process is supplied in the Appendix A. 

### 2.3. Material Characterization

Transmission electron microscopy (TEM) and high-resolution transmission electron microscopy (HRTEM) were performed on a JEM-1200 EX (JEOL, Tokyo, Japan) field-emission transmission electron microscope, operating at an accelerating voltage of 100 kV. X-ray diffraction (XRD) patterns were recorded on a powder X-ray diffractometer (D/MAX-RB, Tokyo, Japan) measured with CuKα radiation (λ = 0.15418 nm) in the 2θ range from 10° to 90°. Fourier-transform infrared spectroscopy (FT-IR) was obtained on a spectrometer (NICO- LET5700, ThermoFisher Scientific, Waltham, MA, USA) by the KBr tablet method. The spectral scanning range was 400–4000 cm^−1^ at a resolution of 4 cm^−1^. X-ray photoelectron spectroscopy (XPS) analysis was performed on an ESCALAB 250 electronic spectrometer (ThermoFisher Scientific, Waltham, MA, USA). UV-vis spectra of N-CQDs were recorded using a spectrophotometer (TU-19, Beijing General Analysis Instrument Co., Ltd., Beijing, China) scanning in the range of 200–500 nm. Fluorescence spectra were obtained with a Fluoromax–4 Spectrofluorometer (HORIBA, Kyoto, Japan).

### 2.4. Detections of Fe^3+^

The selectivity to Fe^3+^ ions was measured by adding different metal ion stock solutions in the same way. Typically, the prepared N-CQDs solutions were diluted 10 times using deionized water; then, 150 μL of 10 mM of each metal ion (Cd^2+^, Mg^2+^, Fe^3+^, Fe^2+^, Cu^2+^, Pb^2+^, Ni^2+^, Ag^+^, Zn^2+^, Al^3+^, Na^+^, Co^2+^, Ca^2+^, Ba^2+^, Hg^2+^) was added to 3 mL of diluted N-CQD solution. The fluorescence spectra were recorded under excitation at 330 nm. A slit width of 5 nm was used for both excitation and emission. The dose-dependent response of N-CQDs to Fe^3+^ was further investigated. Various amounts of Fe^3+^ were added to 3 mL of the diluted N-CQD solution, and the emission intensity of the N-CQDs at 400 nm was recorded under 330 nm excitation. The same volume without Fe^3+^ was applied as the negative control. 

### 2.5. Real Sample Analysis

Analysis of real samples was used to test the practicality of N-CQDs in detecting Fe^3+^ ions in water. Tap water was collected from our lab and lake water was obtained from the lake at Qingdao University. The amount of Fe^3+^ was estimated using the standard addition method. For recovery studies, known concentrations of Fe^3+^ solution were added to the samples and the total iron concentrations were then determined at the same condition.

## 3. Results and Discussion

### 3.1. Preparation and Characterization of N-CQDs

In the present study, water-soluble nitrogen-doped carbon quantum dots (N-CQDs) were prepared by the hydrothermal method with chitosan as both carbon and nitrogen sources (Scheme 1). The as-prepared N-CQDs were well-dispersed yellow solution, which exhibited strong blue fluorescence under UV irradiation (365 nm). The N-CQDs contain a large number of nitrogen-containing (-NH_2_) and oxygen-containing (-OH and -COOH) groups, which can strongly interact with analytes [32]. When iron ions were added to the carbon dot solution, the fluorescence was quenched under UV irradiation (365 nm), which can be discerned by the naked eye. Therefore, a fluorescence sensor based on N-CQDs was constructed and its sensing properties were investigated in detail.

The morphology of the prepared N-CQDs was characterized by TEM and HRTEM. As shown in Figure 1a, the N-CQDs were well-distributed and uniform in size. The size distribution showed a Gaussian fitting curve and a narrow distribution range of 1–4 nm with an average size of about 2 nm (Figure 1b). The HRTEM image of N-CQDs showed well-resolved lattice fringes with an average in-plane lattice spacing of 0.213 nm (Figure 1c), which is very close to the (110) diffraction facets of graphite. This implies that the NCQDs possess a graphite-like structure. The XRD pattern of N-CQDs is shown in Figure 1d. A broad diffraction peak of about 23° (d_002_ = 0.34 nm) and a weak peak at 43.4° (d_100_ = 0.21 nm) were observed in the N-CQDs, revealing an amorphous carbon phase and partial graphitization of the N-CQDs, which are attributed to the existence of abundant functional groups and the graphitization of carbon atoms, respectively. These results further confirm that a graphite structure exists in the carbon core of N-CQDs.

The structure and composition of N-CQDs were investigated by FT-IR spectroscopy (Figure 2). For the pure chitosan powder, the obvious peaks at 3459 cm^−1^ were ascribed to the O–H and N–H stretching vibration, and the absorption bands at 2956 cm^−1^ were attributed to the C–H stretching vibration. The absorption band at 1572 cm^−1^ was assigned to the N–H bending vibration of the primary amino group. The peak at 1441 cm^−1^ was due to C–N stretching vibrations. The peaks at 1638 and 1324 cm^−1^ were ascribed to the vibrations of C=O and C–O bonds, respectively. Compared to chitosan, N-CQDs showed an apparent decrease in the adsorption of O–H and N–H stretching vibrations at 3462 cm^−1^, and C–H vibrations at 1130 cm^−1^, related to the pyranose, almost disappeared, indicating the degradation of the chitosan chain and decomposition of the pyranose ring through dehydration. Additionally, FT-IR analysis confirmed the presence of amine, hydroxyl, and carbonyl functionality in the surface of N-CQDs. 

XPS was also used to characterize the N-CQDs. As shown in Figure 3, the as-prepared N-CQDs contained mainly carbon, nitrogen, and oxygen with their atomic percentages of 72.3%, 6.32%, and 21.38%, respectively (Figure 3a). Furthermore, the high-resolution spectrum of C1s could be deconvoluted into three main peaks at 284.4, 285.2, and 286.5 eV, which could be attributed to sp^2^ carbon (C–C/C=C), sp^3^ carbon C–N, and C–O [33], respectively. The characteristic peak of C–C sp^2^ appearing at 284.4 eV indicates that abundant graphite structures should be contained in the obtained N-CQDs (Figure 3b). The high-resolution spectrum for N1s exhibited two peaks at 399.2 and 400.3 eV, which could be assigned to C–N–C and N–H (Figure 3c), respectively [34]. The high-resolution spectrum for O1s showed a symmetrically shaped peak with a binding energy of 532.1 eV, indicating that the environment of O atoms is uniform in N-CQDs. These data further confirm that the N atom has been successfully doped into CQDs, and functional groups such as C–N, C=C, O–H, and N–H coexisted on the CQD surface.

### 3.2. Optical Properties of N-CQDs

As shown in Figure 4a, the optical properties of the N-CQDs showed a strong absorption band centered at 285 nm, which is attributed to the n–π* transition of C=O, caused by the trapping of the excited state energy of the surface states, which can lead to strong fluorescence. Accordingly, the maximum excitation wavelength was detected at 330 nm. When excited at 330 nm, the N-CQDs exhibited a strong fluorescence emission peak at 410 nm. In addition, in this work, highly fluorescent N-CQDs were fabricated by the hydrothermal carbonization of chitosan. The optimum carbonization time was investigated, as elaborated in Appendix A. The fluorescence intensity increased with the increase in carbonization time. After six hours of carbonization, the chitosan solution turned light yellow in color, while the solution displayed strong blue luminescence under the UV light lamp with excitation at 365 nm. Here, the N-CQDs with carbonization for 24 h were selected for subsequent experiments. Moreover, detailed emission fluorescence spectra of the N-CQDs with different excitation wavelengths were also investigated. Apparently, the emission wavelength of N-CQDs was nearly excitation-dependent, and the maximum emission wavelength shifted from 400 to 480 nm when the excitation wavelength was increased gradually from 300 to 390 nm (Figure 4b), which may be determined by the distribution of different sizes and multi-surface emission sites of N-CQDs [33,35]. Moreover, with the increase in excitation wavelength, the fluorescence intensity of N-CQDs first increased and then decreased, accompanied by a red shift, reaching its maximum at the excitation wavelength of 330 nm. Some works on carbon dots also indicate that the optical properties of carbon dots possibly correspond to fluorescent surface groups [36,37,38]. Additionally, it should be pointed out that the prepared N-CQDs exhibit excellent water solubility and high stability at room temperature. Even after three months of storage in air, no floating or precipitated nanoparticles were observed, and there was no significant loss of photobleaching after 3 h of exposure to a 365 nm ultraviolet lamp, indicating their advantages in potential applications. 

### 3.3. Detection of Metal Ions

To evaluate the selectivity of N-CQDs as a fluorescent probe for Fe^3+^ detection, 150 μL of different metal ions was added to 3 mL N-CQD solution, each at a fixed concentration of 500 µM, and their effects on the fluorescent intensity of N-CQDs were recorded after metal ion addition at room temperature. Figure 5a,b present different quenching effects with various metal ions. Among them, Fe^2+^, Cu^2+^, Ba^2+^, Hg^2+^, Zn^2+^, Ni^2+^, Mg^2+^, Co^2+^, Pb^2+^, Ca^2+^, Al^3+^, Cd^2+^, and Na^+^ metal ions showed a slight change in fluorescence intensity. However, compared to the other metal ions, Fe^3+^ ions caused remarkable fluorescence quenching of N-CQDs.

In other interference tests, 100 μM of Fe^3+^ alone (black bars in Figure 5c) and the mixtures of 100 μM of Fe^3+^ and 100 μM of the above-mentioned metal ions (red bars in Figure 5c) were added into the N-CQD aqueous solution, and the quenching effects of Fe^3+^ and the mixtures of Fe^3+^ and M^n+^ on the fluorescence of the N-CQDs were then examined. The results show that the coexistence of other ions had little effect on the fluorescence quenching of Fe^3+^ to N-CQDs and could even be neglected. These results demonstrate that the N-CQD sensor was insensitive to other metal ions but selective to Fe^3+^ in the mixtures. In addition, the effects of various metal ions on the UV-vis absorption spectrum of N-CQDs were also investigated. As shown in Figure 5d, the UV-vis absorption of N-CQDs changed significantly after adding 100 μM Fe^3+^, and the absorption at 285 nm disappeared. However, the absorption spectra remained almost unchanged after adding the same concentration of other metal ions. The possible reason is that the presence of Fe^3+^ can affect the surface states of quantum dots. The above results strongly prove that the N-CQD sensor has a strong selectivity for Fe^3+^ sensing.

The experimental results indicate that Fe^3+^ could effectively quench the fluorescence of N-CQDs, implying the potential of N-CQDs as a fluorescence probe for the detection of Fe^3+^. In order to obtain the optimal response signal, before quantitative analysis of Fe^3+^, the detection conditions, including pH and reaction time, were investigated. Appendix A shows the fluorescence intensity of N- CQDs at different pH values with 500 µM of Fe^3+^ ion. As can be seen from the figure, when the pH was 2, the fluorescence intensity of N-CQDs was almost unchanged. As the pH value increased from 3.0 to 11.0, the fluorescence quenching efficiency increased rapidly and reached a maximum under pH = 7.0. From the K_SP_ calculation of Fe(OH)_3_, when the pH is about 2, Fe(OH)_3_ precipitation begins to appear in the water solution. Fe^3+^ exists in hydroxide in the alkaline solution, which probably goes against the accurate detection of Fe^3+^ [39]. However, in our reaction system, it is a solution of carbon quantum dots, not pure water solution, and when iron ions are added to the solution, the iron ions probably react with the N-CQDs. Therefore, there is no obvious precipitation of Fe(OH)_3_ even under alkaline conditions. The result clearly indicates that the probe N-CQDs can be used within a broad range of pH values (3.0−11.0). Appendix A shows that the fluorescence of N-CQDs was quenched rapidly after the addition of Fe^3+^ ions within 2 min. In the next experiments, 5 min was selected as the optimum reaction time to obtain stable and accurate experimental results.

Under the optimum conditions, the sensitivity of N-CQDs to Fe^3+^ determination was measured. The fluorescence intensity of N-CQDs at 410 nm gradually decreased with the increase in the Fe^3+^ concentration from 0 to 500 μM (Figure 6a). More importantly, there was a good linear relationship between the fluorescence quenching efficiency (F_0_ – F)/F_0_ and the Fe^3+^ ion concentration within the range of 1–200 μM (Figure 6b). The linear regression equation is (*F*_0_–*F*)/*F*_0_ = 0.002*C* + 0.0585, *R*^2^ = 0.9969, where *F*_0_ and *F* are the fluorescence intensity of N-CQDs before and after the addition of Fe^3+^, respectively. In addition, the detection limit was estimated to be 0.15 μM at a signal-to-noise ratio of 3 [40], which is much lower than recently published assays based on CDs from different precursors (Table 1). Therefore, the N-CQDs could potentially be used as a sensor for Fe^3+^ ion detection. In addition, the detection limit and linear range of this proposed method were compared to other reported CQD-based sensors. As shown in Table 1, this method provides a lower detection limit and a comparable wide linear range. Therefore, we believe that our present method can be applied to the detection of Fe^3+^ ions.

### 3.4. Fluorescence Quenching Mechanism of the N-CQD-Fe^3+^ System

Most N-CQDs are sensitive to metal ions because of their external abundant carboxyl and hydroxyl groups. When Fe^3+^ ions were added to the prepared N-CQD aqueous solution, the fluorescence emission intensity was significantly quenched. This strong quenching effect may originate from the formation of N-CQD-Fe^3+^ complexes [47]. Based on previous studies, a reasonable mechanism for the fluorescence quenching of N-CQD is the intramolecular photoinduced electron transfer process from excited N-CQD to Fe^3+^ ions (Scheme 2). When Fe^3+^ was added, the electron-deficient Fe^3+^ complexes with functional groups on the surface of the N-CQD (for example, carboxyl groups and amino groups) lead to the splitting of the d-orbital of Fe^3+^. Therefore, electrons in the excited state of N-CQD were partially transferred to the d-orbital of Fe^3+^. Electron transition in the radiation forms (FL emission) was consequently restrained, resulting in fluorescence quenching [48]. To further clarify the mechanism of fluorescence quenching, the N-CQD-Fe^3+^ complex was characterized by XPS. As shown in Figure 7, the Fe 2p spectrum presented two peaks at 724.6 and 711.2 eV, corresponding to Fe 2p_1/2_ and Fe 2p_3/2_ energy levels of the Fe atom, respectively, suggesting the presence of metallic Fe. Further, as shown in Figure 5d, the absorption spectrum of N-CQDs changed remarkably after adding Fe^3+^, the absorption at 285 nm disappeared, and a new absorption peak appeared near 370 nm. This clearly shows that the surface states of the N-CQDs were modified due to the combination of Fe^3+^ ions. These results indicate that the presence of Fe^3+^ could lead to the formation of N-CQD-Fe^3+^ complexes.

### 3.5. Application to Real Samples

To investigate the practical feasibility of N-CQD colorimetric probes, it was applied to the determination of Fe^3+^ in real tap water and lake water samples. The collected water sample was filtered through a 0.45 μm membrane to remove insoluble materials before detection. The standard addition method was used to determine the unknown concentration of Fe^3+^ in the water samples. In addition, recovery assays for different concentrations of Fe^3+^ were performed. As can be seen from Appendix A, the spiked recoveries were in the range of 96.8–101.9% and the relative standard deviations (RSDs) were less than 3.75%, which indicates that the sensing system showed excellent applicability for real sample analysis.

## 4. Conclusions

In summary, we have reported a simple, economic, and green method to produce water-soluble nitrogen-doped carbon quantum dots (N-CQDs) with chitosan as both carbon and nitrogen sources. The as-prepared N-CQDs exhibited high water solubility, stability, and excitation-dependent behavior. The average diameter of the N-CQDs was about 2 nm. Moreover, based on the dramatic fluorescence quenching capacity of Fe^3+^, the N-CQDs presented outstanding selectivity and sensitivity and were successfully applied for the quantitative detection of Fe^3+^ in aqueous solutions with a linear detection range of 0–500 μM and detection limit of 0.15 μM. This method is simple, rapid, and efficient, which can be used for the detection of Fe^3+^.

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
