# Peer review of "Facile Synthesis of Nitrogen-Doped Carbon Quantum Dots with Chitosan for Fluorescent Detection of Fe3+"

_polymers, 2019, doi:10.3390/polym11111731_

Round 1

Reviewer 1 Report

In this manuscript, the authors have developed a facile, economical and one-step hydrothermal method to prepare highly luminescent nitrogen-doped carbon quantum dots (N-CQDs) with chitosan as both carbon and nitrogen sources. The resulting N-CQDs exhibit excitation wavelength-dependent fluorescence with the maximum excitation and emission at 330 nm and 410 nm, respectively, and can be used as a fluorescent sensor for Fe3+ ion sensitive detection. The manuscript can be published after minor revisions.

1 The abbreviations, such as FT-IR, UV–vis, should be used the same format throughout the manuscript.

2 Some typos or other errors should be corrected, here are some of them:

Line 9

A facile, economical and and one-step hydrothermal method

A facile, economical and one-step hydrothermal method

Lines 11-13

The as-prepared N-CQDs have an average size of 2 nm and exhibit excitation wavelength-dependent fluorescence with the maximum emission and excitation at 330 nm and 410 nm, respectively.

The as-prepared N-CQDs have an average size of 2 nm and exhibit excitation wavelength-dependent fluorescence with the maximum excitation and emission at 330 nm and 410 nm, respectively.

Line 46

sensi- tivity

sensitivity

Line 52

Chitosan

chitosan

Line 54

it has natural chelating properties that can be combined and removes heavy

it has natural chelating properties that can be combined and used to remove heavy

Lines 64 and 65, 85 and 86

All metal salts (CdCl 2 、MgCl 2 、FeCl 3 、FeCl 2 、CuCl 2 、 64

Pb(NO 3 ) 2 、Ni(NO 3 ) 2 、AgNO 3 、ZnCl 2 、AlCl 3 、NaCl、Co(NO 3 ) 2 、CaCl 2 、BaCl 2 、HgCl 2 )

15 μL 10 mM of each metal ion (Cd 2+ 、Mg 2+ 、Fe 3+ 、Fe 2+ 、Cu 2+ 、Pb 2+ 、Ni 2+ 、Ag + 、Zn 2+ 、Al 3+ 、Na + 85

、Co 2+ 、Ca 2+ 、Ba 2+ 、Hg 2+ )

Please use “,” instead of “、”

Line 119

Inset of (c) showned the HRTEM image

Inset of (c) showed the HRTEM image

Lines 124 and 125

the peaks at 1638 cm-1 and 1324 cm -1 were ascribed to the vibrations of C=O and C-O bonds respectively. compared to chitosan, N-CQDs

The peaks at 1638 cm-1 and 1324 cm -1 were ascribed to the vibrations of C=O and C-O bonds respectively. Compared to chitosan, N-CQDs

Line 136

XPS was also used to characterize the N–CQDs,

XPS was also used to characterize the N–CQDs.

Line 140

the characteristic peak of C–C sp 2 appearing at

The characteristic peak of C–C sp 2 appearing at

Line 142

High resolution spectrum for N1s exhibited two peaks at 399.2 eV, 400.3 eV,

High resolution spectrum for N1s exhibited two peaks at 399.2 eV and 400.3 eV,

Line 145

These data further confirmed that the N atom has been successfullu

These data further confirmed that the N atom has been successfully

Line 151

Accordingly, The maximum

Accordingly, the maximum

Line 173

Fluorescence excitation (λex= 330 nm, green line) and emission (λem= 410 nm, blue line) spectra of N-CQDs

fluorescence excitation (λem= 410 nm, green line) and emission (λex= 330 nm, blue line) spectra of N-CQDs

Line 181

Pb 2+ ,Ca 2+ , Al 3+ , Cd 2+ , Na + metal

Pb2+ , Ca2+ , Al3+ , Cd2+ and Na+ metal

Line 239

Fluorescence emission intensity was significantly quenched.

fluorescence emission intensity was significantly quenched.

3 For the references, the authors should always use the same style. Please check them and revise some of them.

Author Response

Dear reviewer,

Thank you for your positive comments on my article (MS Number: Polymers-600930)! Those comments are all valuable and very helpful for revising and improving our paper, as well as the important guiding significance to our research. We have revised the manuscript according to your detailed suggestions. Enclosed please find the responses to the reviewers. Thank you very much for all your help and we are looking forward to hearing from you. If you have any question about this paper, please don’t hesitate to let us know.

Sincerely yours,
Xihui Zhao
E-mail address: [email protected]

Reviewer 2 Report

Lines 12-13: The „emission and excitation at 330 nm and 410 nm, respectively.“ must be corrected.

Lines 16-19: The Fe+3 has just been measured in tap water as a real sample which is not a complex real sample. The new developed analytical method must be tested for target determination in more contaminated water samples e.g. river water or wastewater samples.

Lines 35-39: In these lines, importance of Fe+3 determination is highlighted due to medical reasons. However, the method was developed for its determination in water samples. Please provide environmetntal-related infomation confirming importance of Fe+3 determination.

Lines 42-43:  Due to my interest to see, how could be e.g. an electrochemical method more complicated than your proposed method, I have checked the refernces in these lines. SURPRISINGLY, non of them is related to the methods which are mentioned for them. WHY? I don‘t have time to check the other references. However, I assume that could be the points for other references.

Line 56: Please provide suitable references after „… the removal of heavy metal.“.

Lines 85-86: Using provided information, the final concentrations could be cacluated at 50 µM. However, it is reported in figure 5: 500 µM and in line 178: 500 mM  . Which one is correct?

Line 95+section 3.2.+Figure4b+ Figure S1.b: Why is excitation at 365 nm used when optimied one is 330 nm?

Lines 100-102: The sentence: „Therefore, a ratiometric  fluorescence sensor based on N-CQDs was constructed and its recognition/sensing properties were investigated in detail [28].“, what is the relation of this sentence to the text?

Line 106: Did author use the same instrument for TEM and HRTEM?

Figure 4a: The graph shows the absorbance on the left and fluorescence on the right. How about excitation (green one)?

Figure 5d: The picture is not clear and has a mistake.

Lines 208-210: Please provide an explanation for pH evaluation. What will happen for Fe+3 at alkaline conditions. Is it soluble?

Author Response

(The authors gave the same response as above.)

Reviewer 3 Report

In the current manuscript, the authors report the results of synthesis of water soluble nitrogen-doped carbon quantum dots.The as-prepared particles exhibited high water solubility, stability and excitation-dependent behavior. The samples show high sensitivity of fluorescence intensity on presence of Fe3+ in water. Technically, the manuscript contains enough information to reproduce the results;the presentation of the data is in general clear, although some minor issues can be improved.

However, my main concern is related to the lack of novelty of the current work. Based on the state-of-the-art of carbon dots research, simplicity of nanoparticle synthesis is not enough to trigger the progress in the field. There are numerous reports on Fe3+ sensing using various types of carbon dots, see for instance:

Scientific Reports, 7, 14866 (2017);
Optical Materials, 89, 2019, 92-99;
Analyst 2018,143, 5812-5821; and many more...

The authors can think of using more advanced ways of sensing that simply fluorescence intensity. It may be change of excited state lifetime or spectral shifts. The latter parameters allow one to do more quantitative measurements that re not sensitive to change of concentration. Based on the above arguments, unfortunately, I cannot recommend this manuscript for publication.

Author Response

Manuscript ID: Polymers-600930

Title: Facile synthesis of nitrogen-doped carbon quantum dot with chitosan for fluorescent detection of Fe3+

Journal:Polymers

Correspondence Author: Xihui Zhao

Dear reviewer,

Thank you for your comments on my article (MS Number: Polymers-600930)! Those comments are all valuable and very helpful for revising and improving our paper, as well as the important guiding significance to our research. Enclosed please find the responses to the reviewers. Thank you very much for all your help and we are looking forward to hearing from you. If you have any question about this paper, please don’t hesitate to let us know.

Sincerely yours,
Xihui Zhao
E-mail address: [email protected]

Round 2

Reviewer 2 Report

Comment 2: Lines 16-19: The Fe3+ has just been measured in tap water as a real sample which is not a complex real sample. The new developed analytical method must be tested for target determination in more contaminated water samples e.g. river water or wastewater samples.

[Response]: Thanks for the reviewer’s kind advice. We have tested the Fe3+ in the wastewater samples and added the results to the Table S1. (Lines 17-18)

Comment to the [Response]: What is the evaluated wastewater? Influent or effluent? From which WWTP are the samples taken? Detected target in tap water and wastewater samples are nearly in the same range (0.5 and 0.57 µM), for me it is weird.

Comment 3: Lines 35-39: In these lines, importance of Fe3+ determination is highlighted due to medical reasons. However, the method was developed for its determination in water samples. Please provide environmental-related information confirming importance of Fe3+ determination.

[Response]: Thanks for the reviewer’s kind advice. We have added relevant content in the manuscript and marked in red. (Lines 35-39 and Line 42-44)

Comment to the [Response]: Again, the used reference, [12], does not provide what was requested. In the reference [12], authors developed a electrochemical sensor for determination of Pb(II), Hg(II), Cu(II) and Cd(II). Where is Iron?

Comment 5: Line 56: Please provide suitable references after “the removal of heavy metal”.

[Response]: Thanks for the reviewer’s kind advice. We have added suitable references in the manuscript and marked in red. (Lines 60)

Comment to the [Response]: In the manuscript: “Therefore, there is growing interest to explore the use of chitosan in wastewater treatment for the removal of heavy metal [28-30]”.

I have just checked the reference [28]: in the first line of the abstract is: “We have described a simple and reliable colorimetric method for the sensing of biothiols such as cysteine, homocysteine, and glutathione in biological samples.” Where is wastewater???Where is heavy metal ???

Comment 8: Lines 100-102: The sentence: “Therefore, a ratiometric fluorescence sensor based on N-CQDs was constructed and its recognition/sensing properties were investigated in detail [28]”. What is the relation of this sentence to the text?

[Response]: Thanks for the reviewer’s kind advice. The reference has been modified.

Comment to the [Response]: The comment 8 is still not answered.

Comment 12: Lines 208-210: Please provide an explanation for pH evaluation. What will happen for Fe3+ at alkaline conditions. Is it soluble?

[Response]: Thanks for the reviewer’s kind advice. We supplemented the content of the pH evaluation in our manuscript and marked in red. (Lines 214-220)

Comment to the [Response]: I did not ask authors to describe Figure S2 in text. I needed a scientific explanation for fig. S2. Ksp for Fe(OH)3 is about 2.79E-39. Can we have 500µM of fe+3 in water samples at pH above 7??

Author Response

(The authors gave the same response as above.)

Reviewer 3 Report

In the revised manuscript, the authors have addressed minor issues that were present in the initial version. Unfortunately, I do not see any improvement of the key point, namely the lack of novelty of the work. This work shows certain minor improvements (or more specifically, modifications) of the existing methods, which will potentially be used by specialists in this field. Therefore, given that the authors have no possibility to perform more advanced experiments within a reasonably limited time, I am ready to accept that this manuscript can be published in the current form rather as a technical/methodological note.

Author Response

Dear reviewer,

Thank you for your positive comments on my article (MS Number: Polymers-600930)! Those comments are all valuable and very helpful for revising and improving our paper, as well as the important guiding significance to our research. Enclosed please find the responses to the reviewers. Thank you very much for all your help and we are looking forward to hearing from you. If you have any question about this paper, please don’t hesitate to let us know.

Sincerely yours,
Xihui Zhao
E-mail address: [email protected]

Round 3

Reviewer 2 Report

Comment 1:

Comment 2: Lines 16-19: The Fe3+ has just been measured in tap water as a real sample which is not a complex real sample. The new developed analytical method must be tested for target determination in more contaminated water samples e.g. river water or wastewater samples.[Response]: Thanks for the reviewer’s kind advice. We have tested the Fe3+ in the wastewater samples and added the results to the Table S1. (Lines 17-18) 

Comment to the [Response]: What is the evaluated wastewater? Influent or effluent? From which WWTP are the samples taken? Detected target in tap water and wastewater samples are nearly in the same range (0.5 and 0.57 µM), for me it is weird.

[Response]: Thanks for your taking the time to offer us the comments and insights related to the paper. Tap water was collected from our lab and wastewater was obtained from the lake on campus. The details have been supplemented in section 2.5 and 3.5 in the manuscript. (Line 98-103 and Line 270-278)

Comment to the [Response]: Water sample from lake is not wastewater, it is lake water sample. Wastewater samples are highly contaminated influent or effluent water samples from a waste water treatment plant. Please correct them in the manuscript.

-----------------------------------------------------

Comment 5:

Comment 12: Lines 208-210: Please provide an explanation for pH evaluation. What will happen for Fe3+ at alkaline conditions. Is it soluble?

[Response]: Thanks for the reviewer’s kind advice. We supplemented the content of the pH evaluation in our manuscript and marked in red. (Lines 214-220)

Comment to the [Response]: I did not ask authors to describe Figure S2 in text. I needed a scientific explanation for fig. S2. Ksp for Fe(OH)3 is about 2.79E-39. Can we have 500µM of fe+3 in water samples at pH above 7??

[Response]: Thanks for your taking the time to offer us the comments and insights related to the paper. Iron ions have a hydrolysis equilibrium in water, and precipitate begins to occur when pH=2. But in our system, here is a carbon quantum dots solution, not pure water solution. When iron ions are added to the solution, the iron ions will react with the N-CQDs.

Comment to the [Response]: Please I) provide all these information in the manuscript and II) provide suitable reference for this claim.

Author Response

Dear reviewer,

Thank you for your positive comments on my article (MS Number: Polymers-600930)! Those comments are all valuable and very helpful for revising and improving our paper, as well as the important guiding significance to our research. We have revised the manuscript according to your detailed suggestions. Enclosed please find the responses to the reviewer. Thank you very much for all your help and we are looking forward to hearing from you. If you have any question about this paper, please don’t hesitate to let us know.

Sincerely yours,
Xihui Zhao
E-mail address: [email protected]
